# Light modulates task-dependent thalamo-cortical connectivity during an auditory attentional task

Ilenia Paparella [1], Islay Campbell[1], Roya Sharifpour[1], Elise Beckers[1,2], Alexandre Berger[1,3,4], Jose Fermin Balda Aizpurua[1], Ekaterina Koshmanova[1], Nasrin Mortazavi[1], Puneet Talwar [1], Christian Degueldre[1], Laurent Lamalle[1], Siya Sherif[1], Christophe Phillips [1], Pierre Maquet [1,5] & Gilles Vandewalle [1✉]

Exposure to blue wavelength light stimulates alertness and performance by modulating a widespread set of task-dependent cortical and subcortical areas. How light affects the crosstalk between brain areas to trigger this stimulating effect is not established. Here we record the brain activity of 19 healthy young participants (24.05±2.63; 12 women) while they complete an auditory attentional task in darkness or under an active (blue-enriched) or a control (orange) light, in an ultra-high-field 7 Tesla MRI scanner. We test if light modulates the effective connectivity between an area of the posterior associative thalamus, encompassing the pulvinar, and the intraparietal sulcus (IPS), key areas in the regulation of attention. We find that only the blue-enriched light strengthens the connection from the posterior thalamus to the IPS. To the best of our knowledge, our results provide the first empirical data supporting that blue wavelength light affects ongoing non-visual cognitive activity by modulating task-dependent information flow from subcortical to cortical areas.

[1] GIGA-Cyclotron Research Centre-In Vivo Imaging, University of Liège, 4000 Liège, Belgium. [2] Alzheimer Centre Limburg, School for Mental Health and Neuroscience, Faculty of Health, Medicine and Life Sciences, Maastricht University, 6229 ET Maastricht, The Netherlands. [3] Institute of Neuroscience (IoNS), Université Catholique de Louvain (UCLouvain), 1200 Brussels, Belgium. [4] Synergia Medical SA, 1435 Mont-Saint-Guibert, Belgium. [5] Neurology Department, CHU de Liège, 4000 Liège, Belgium. ✉email: gilles.vandewalle@uliege.be

Light provides more than just visual information through non-classical photoreception (also referred to as non-image forming—NIF effects)[1–3]. Non-visual responses affect physiological processes such as circadian entrainment, heart rate, body temperature, hormone secretion, pupil light reflex, alertness, and sleep propensity. These responses are mediated to a large extend by intrinsically photosensitive retinal ganglion cells (ipRGCs) that constitute a distinct class of photoreceptors expressing the photopigment melanopsin, most sensitive to blue wavelength light at around 480 nm[4,5]. Animal and human studies show that ipRGC melanopsin-driven intrinsic responses concur with inputs from rods and cones to increase the sensitivity in the blue portion of the light spectrum (460–480 nm)[1,2,5,6].

Light has a direct activating effect on cognition and acutely increases performance following the onset of light[1,2,7]. Positron emission tomography and magnetic resonance imaging (MRI) studies showed that light modulates the activity of a widespread set of cortical and subcortical areas, whose distribution partly depends on the ongoing task (see ref. [7] for a review). However, the neural processes underlying light-induced modulation of non-visual cognitive performance are not fully established. IpRGCs project to several brain areas including subcortical hypothalamic, thalamic and brainstem structures involved in attention and alertness regulation[8]. These areas were reported to respond to short light exposure in humans, suggesting they are in a strategic position in forwarding non-visual light information to the cortex[7,9]. The pulvinar, a posterior thalamic nuclei, is one of the structures more consistently activated in response to light in human fMRI studies investigating the impact of light on ongoing non-visual cognitive activity[7,10–12]. It is involved in attention control and, together with adjacent multimodal associative nuclei (e.g., the dorsomedial nucleus), modulates ongoing cortical activity through the recurrent thalamo-cortical loops[13,14]. It is also connected with the suprachiasmatic nucleus (SCN) of the hypothalamus, the principal circadian clock[15] that, in rodents, receives strong and direct input from ipRGCs[8]. Importantly, changes in thalamic activity were found to be directly related to the subjective improvement of alertness induced by light exposure[12]. The posterior thalamus could therefore be an essential structure through which light affects the information flow in the brain while completing non-visual cognitive tasks.

Here, we provide empirical support for this assumption. In an ultra-high-field 7 Tesla MRI scanner, we recorded the brain activity of 19 healthy young participants while they completed a purely auditory attentional task known to recruit the posterior thalamus. Participants were alternatively maintained in darkness or exposed to 30s-blocks of active, blue-enriched polychromatic, or control orange monochromatic light, which, importantly, differ in terms of their predicted stimulation of ipRGCs (90 vs. 0.16 melanopic equivalent daylight illuminance -mel EDI- lux). We analyzed how light information modulated the effective connectivity from the dorso-posterior thalamus to the intraparietal sulcus (IPS), a key cortical area for attentional control, during the auditory task. Only the active light was found to strengthen the connectivity between the thalamus and the IPS compared to the darkness and the control light condition. We further observed, in a subset of participants, that the impact of blue-enriched light on effective connectivity was correlated to changes in pupil size, which constitutes another readout of the non-visual impact of light on physiology. Taken together, our findings support that blue wavelength light affects ongoing non-visual cognitive activity by modulating task-dependent information flow from subcortical to cortical areas.

## Results

Participants were asked to perform an oddball task where rare deviant tones were presented randomly among most frequent standard ones. The overall performance was high (mean $0.96 \pm 0.005$), and the reaction times (RTs) to the deviant tones were not influenced by the light condition ($t(19) = 2.10$, $p = 0.47$, Cohen's $d = 0.05$) (Supplementary Fig. 1a, b). This was expected as, although it engages attentional processes, the task is easy and leads to ceiling performance. The behavioral results are nevertheless important as they exclude that behavioral performance differences unspecific to light exposure would significantly bias fMRI results. However, the measures we collected during the protocol does not allow us to exclude that alertness and/or attention were modulated by the short light exposure without affecting performance to the task (see Campbell et al.[16] for an impact of light on evoked pupil responses that are related to attention and alertness).

**Univariate analysis on the response to deviant tones.** We first completed a standard univariate analysis to isolate thalamus and IPS responses to the deviant tones irrespective of the light condition. We subsequently probed their effective connectivity across light conditions. A widespread set of regions responded to deviant tones, in line with prior literature (Fig. 1). We found a bilateral activation of the primary auditory cortex and the visual cortex (V1), coherent with the fact that subjects were performing an acoustic task under different light conditions. We also detected activations in the right cerebellum and the left primary motor cortex (BA4), reflecting the need for a motor response from the participants. We further found bilateral activation of the insula, the anterior IPS (Fig. 1a) and the dorso-posterior thalamus (Fig. 1b), which are implicated in error awareness and salience processing[17], top-down regulation of attention[18] and target detection[19], respectively.

The first principal component of blood oxygen level dependent (BOLD) time series was extracted bilaterally from the thalamus and the IPS to infer their respective neuronal activity. We estimated the effective connectivity among the thalamus and IPS using dynamic causal modelling (DCM)[20]. To provide a visual representation of the anatomical location of the ROIs, we combined them across subjects to create probabilistic maps (Fig. 1a, b and Supplementary Tables 1 and 2 provides the central coordinates of individual ROIs). We further superimposed the probabilistic maps onto an MRI-based parcellation of the thalamus[21] to delineate the thalamic nuclei encompassing our activation (Fig. 1c). Thalamus activation encompassed a large portion of the pulvinar as well as other multimodal associative nuclei: ventral posterolateral (VPl) and ventral lateral posterior (VLp), highly interconnected with the somatosensory[22] and the motor cortex[23]; centromedian (CM) and mediodorsal (MD-Pf), involved in attention and arousal[24].

**Blue-enriched light increases the effective connectivity from the thalamus to IPS.** DCM analyses first indicated that the driving input, the deviant tones, was effective for both the pulvinar and IPS in both hemisphere models (Pp ≥ 0.95) and was excitatory. This is compatible with the fact that neither of the two regions are primary sensory areas such that they can both receive secondary inputs. The baseline effective bidirectional connectivity between the posterior thalamus and the IPS was then estimated without modulation by the light exposures. Analyses yielded significant reciprocal negative influence between both regions, on top of a relatively weak self-inhibitory feedback for the left thalamus and a relatively stronger self-inhibition feedback bilaterally for IPS (Pp ≥ 0.95) (Fig. 2a). Critically, when considering the

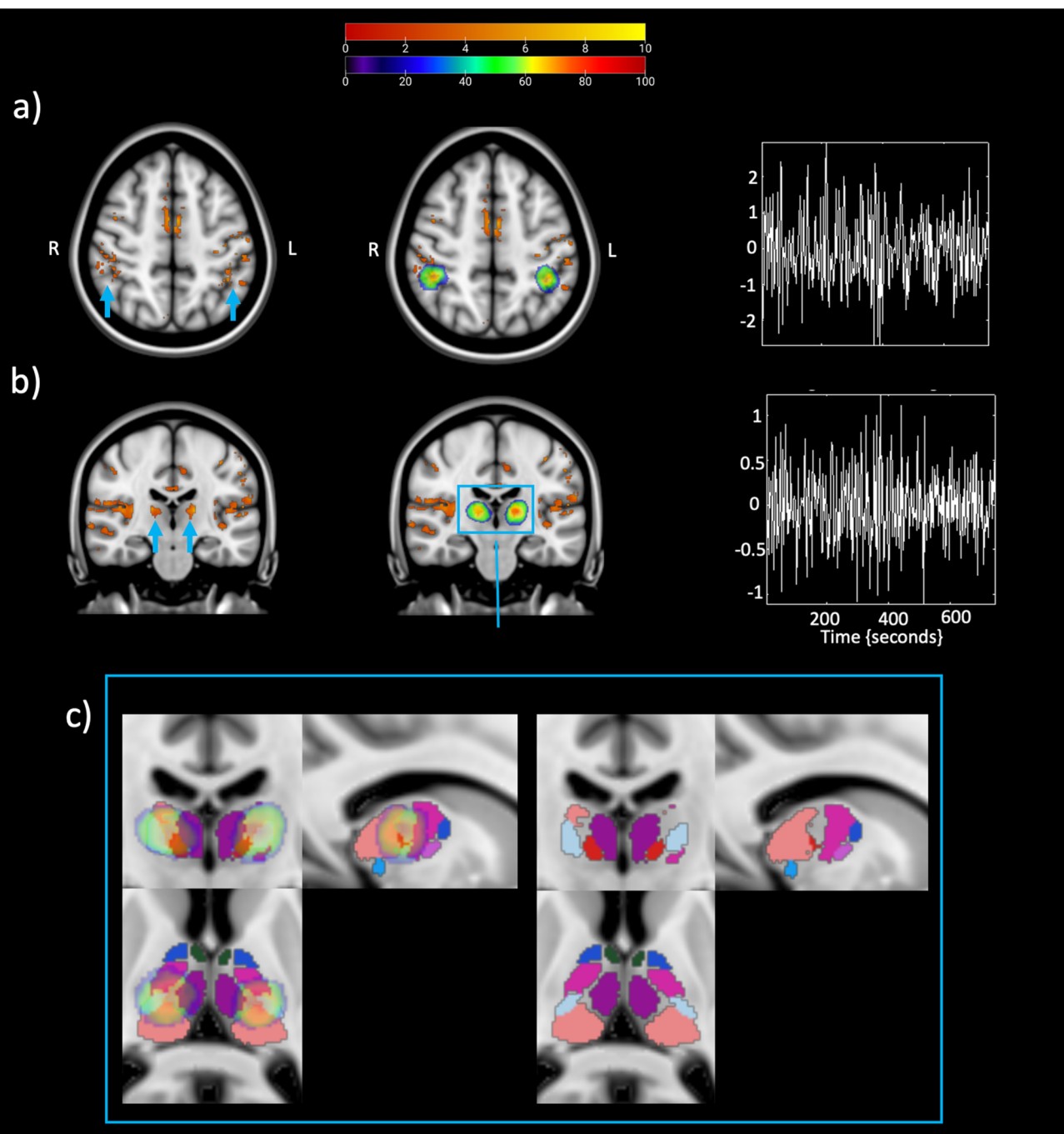

**Fig. 1 Activation and eigenvariate extraction from the two regions of interest (ROIs). a** The intraparietal sulcus (IPS). **b** The thalamus. Left: axial or coronal view showing brain areas activated during the appearance of deviant tones. The upper legend shows the *t*-values associated with the color map. Results are thresholded at *p* < 0.05 FDR-corrected. The blue arrow is pointing at either the bilateral intraparietal sulcus (peak MNI coordinates left hemisphere: [−40 −46 50], $Z_{score} = 4.16$, $P_{FDR-corr} = 0.002$; right hemisphere: [45 −30 51], $Z_{score} = 3.47$, $P_{FDR-corr} = 0.008$) or at the bilateral thalamus (peak MNI coordinates left hemisphere: [−16 −21 11], $Z_{score} = 4.61$, $P_{FDR-corr} = 0.001$; right hemisphere: [17 −20 12], $Z_{score} = 4.45$, $P_{FDR-corr} = 0.002$). Center: probabilistic ROIs across subjects overlapped onto the activation map for both ROIs. The color scale showed in the lower legend represents the proportion of subjects whose ROI included that node: the redder the color the higher the probability that the node is common across the subjects. Right: an example of the adjusted eigenvariate in both ROIs. **c** Thalamic probability maps on parcellation. Left: Zoom in of the thalamic probability maps overlapped onto an MRI-based parcellation of the thalamus[21] in all three views. Right: Parcellation alone to fully show the nuclei encompassing our thalamic activation.

modulatory effect of the active (Fig. 2b) and the control (Fig. 2c) lights, the analyses yielded a single significant modulation of the connection going from the posterior thalamus to the IPS connection only under the active, blue-enriched, light condition which switched the influence of the posterior thalamus on IPS from inhibition to excitation (Pp ≥ 0.95). We then tested whether

this increased effective connectivity under blue-enriched light was related to the pupil reaction to light. This exploratory analysis was conducted on a subset of 11 participants (out of 19) with good pupil data. We computed the averaged modulation exerted by blue-enriched light over the TH-IPS connection (referenced to baseline) across both hemispheres. We also calculated the

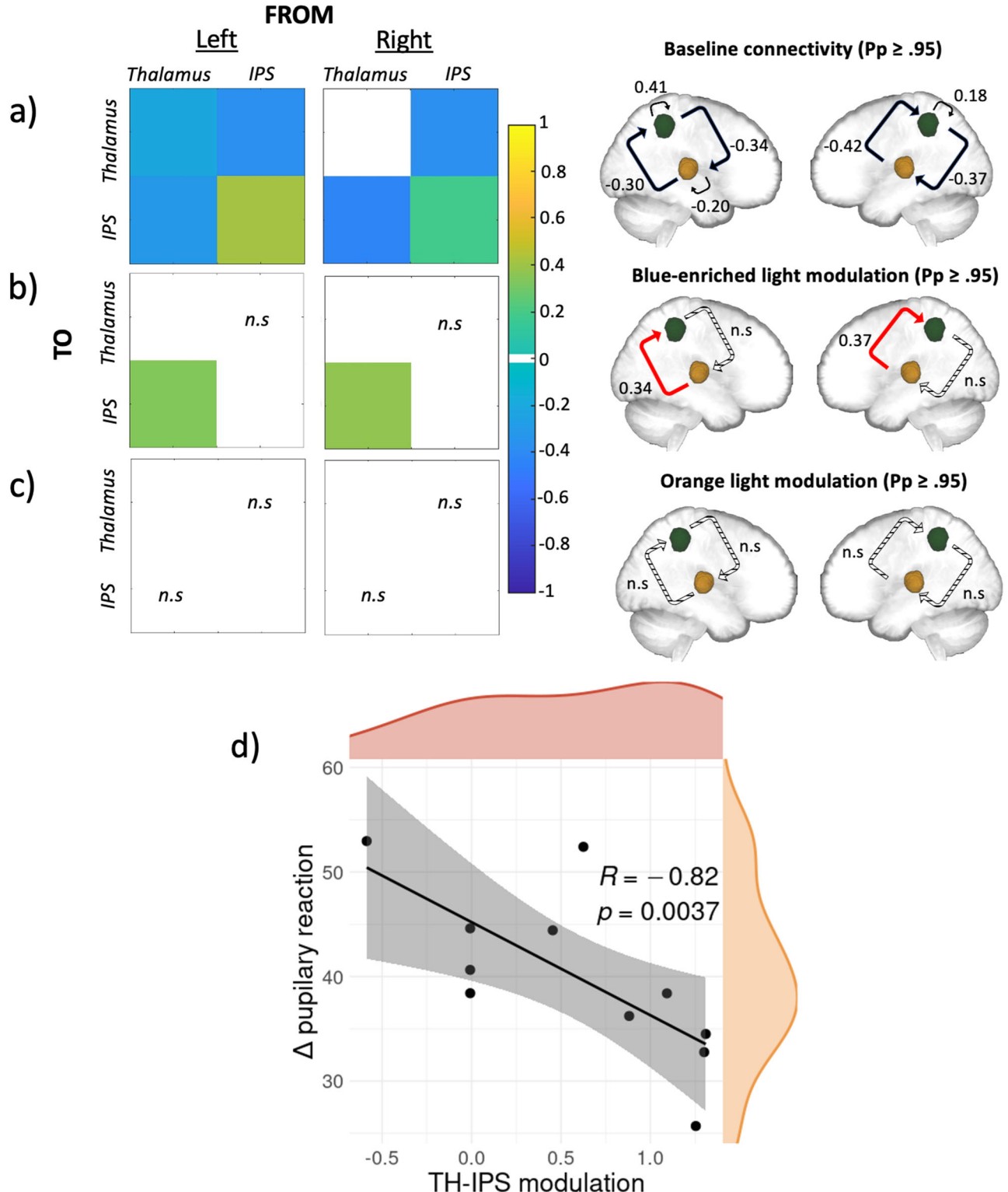

difference in pupil constriction in both lights exposures as detailed in the "Pupillary response" method section. Given the small sample size, we then computed Spearman's correlation, which yielded significant negative correlation ($r[9] = -0.82$; $p = 0.0037$; Fig. 2d).

## Discussion

In this study we tested whether light could affect an ongoing non-visual cognitive task through the modulation of cortical activity by the dorso-posterior thalamus. We asked healthy young participants of both sexes to perform an auditory attentional task which is devoid of visual inputs processing and elicits robust reproducible brain responses, including in attentional-related brain areas such as the parietal cortex[25]. Critically, while performing the task, participants were exposed to blocks of blue-enriched polychromatic (active) light or monochromatic orange (control) light interleaved by periods of complete darkness. We probed the effective connectivity between the thalamus and the

**Fig. 2 Effective connectivity results and their relationship with pupillary response. a–c** PEB results of the DCM analysis at baseline, and under blue-enriched or orange light modulation, respectively. Left: matrices of the effective connectivity either at baseline (**a**) or with modulatory effects exerted by the active blue-enriched (**b**) or the control orange (**c**) light for either the left or the right hemisphere. Right: schematic representation of the corresponding matrices where IPS and the thalamus are showed in green and yellow respectively. On all panels, only suprathreshold parameters are shown (Pp > 0.95) whereas subthreshold parameters are marked as "n.s" (i.e., non-suprathreshold). Connections strengths are represented on a scale from turquoise to yellow, if excitatory, and from light to dark blue if inhibitory. Non modeled direct effects (i.e., whose priors were set to 0) are displayed in white. In the schematic representation, the line patterns denote whether the connection was significantly modulated compared to baseline: solid and dashed lines represent connections that were significantly modulated or not, respectively; red lines denote excitatory connections whereas black ones denote inhibitory connections. **d** Spearman correlation results. Spearman correlation between modulation exerted by blue-enriched light over the TH-IPS connection (referenced to baseline) across both hemispheres and the difference in pupil constriction in both lights exposures ($r[9] = -0.82$; $p = 0.0037$). Density plots for both variables are also provided in orange and red respectively.

IPS and, in line with our hypothesis, we found (1) that the connection from the thalamus to the IPS was selectively modulated by light and (2) that only the blue-enriched light exerted a significant modulation of the information flow along this connection. Interestingly, the modulation exerted by blue-enriched light may be proportional to the concomitant pupil constriction.

The thalamus is a heterogeneous structure that drives nearly all sensory incoming information to the cortex. It is subdivided into distinct nuclei with specific patterns of anatomical connections[14]. These nuclei are classified in relay nuclei which convey information to functionally discrete somatosensory areas; non-specific nuclei, which diffusely project to the cortex and are deemed to be involved in the regulation of alertness; and association nuclei, found in a large part of dorsomedial and posterior nuclei, such as the pulvinar, which integrate sensory information by projecting to association cortices, such as the parietal cortex (see ref. [26] for a review on the thalamic connectivity). Studies on humans and non-humans primates[27] showed that the most consistent structural connection between the parietal cortex, in particular the IPS, and the thalamus passed through the lateral posterior nucleus (e.g., the lateral geniculate nucleus -LGN) and a large portion of the pulvinar. This transfer is most probably monosynaptic and profoundly modulates cortical activity[28].

Based on this, we hypothesized that non-visual effects of light on the posterior thalamus/pulvinar could greatly modulate the activity of top-down attention-related areas, such as IPS[18], resulting in improved alertness and attention to an ongoing auditory attentional task such as the oddball. Previous human fMRI studies identified the posterior thalamus/pulvinar as the area most consistently affected by light exposure during non-visual cognitive tasks[7,10–12]. However, these studies did not specify the directionality of the effect of light on thalamo-cortical connections[12]. Here, we precisely investigate the effects of light on thalamo-cortical effective connectivity. We used a state-of-the-art probabilistic connectivity approach to assess which connection between the thalamus and the IPS was affected by light. We found that only the subcortical to cortical connectivity was significantly affected (posterior probability ≥ 0.95) by light exposure and not by any light but just by the active light condition, i.e., the exposure meant to strongly recruit ipRGCs. In other words, the active light modulated specifically the information flow from thalamic to cortical areas, while the control light, despite otherwise eliciting visual responses, was not affecting our network in any way. Importantly, two models were computed independently over regions of the left or right hemispheres and yielded the same results, providing some cross-validation within our dataset.

Non-visual light information is transferred to the brain through the optic nerve by ipRGCs, which seem to constitute the only channel through which light triggers non-visual responses[6]. A single ipRGC can target up to five different brain areas involved in many distinct light-mediated behaviors. However, the size and the complexity of the axonal arborization are maximally elaborated in the SCN, which is considered to be the primary ipRGCs target[29]. Thus, light could indirectly reach the thalamus via diencephalic afferents from the SCN to the ventral LGN or via the projections from the SCN to the thalamic paraventricular nucleus[30]. Irradiance information could also be transferred to the thalamus through a multisynaptic pathway involving the locus coeruleus (LC)[31], which receives SCN inputs via the dorsomedial hypothalamus (DMH)[32]. The ipRGCs that innervate the SCN also sends collateral projections to the lateral hypothalamic area[29], which contains the cell bodies of orexinergic neurons regulating wakefulness and substantially projects to the para-taenial and the paraventricular nuclei of the thalamus[33]. Besides targeting many hypothalamic nuclei, ipRGCs also send projections to the thalamus in the paraventricular nucleus[34], the intergeniculate leaflet (IGL)[35], corresponding to the human ventral LGN, and the pulvinar[36], which could constitute a direct non-visual pathway from the retina to the thalamus.

Based on our data, we cannot, however, isolate which of these pathways may be involved in the effects of light we detected. In addition, since our light conditions differed both in terms of melanopsin and rod/cone stimulation we cannot separate the contribution of each retinal photoreceptors. While ipRGCs are arguably involved, cones (and even rods[37]) might be involved through their contribution to ipRGC overall responses. Rods and cones could also reach the posterior thalamus/pulvinar through the classical visual pathway. The photic sensory input could first reach the LGN and then the primary visual cortex. This primary sensory area could then send the information back to both the thalamic relay nucleus and to a high order-association nuclei, such as the pulvinar, which could in turn drive alertness in associative cortical regions[38]. Despite being an exploratory analysis on a subset of 11 participants, the fact that the TH-IPS modulation exerted by blue-enriched light was correlated to the changes in pupil constriction induced by blue-enriched, may support that our connectivity findings consist of a non-visual response to light. Pupil response to light constitute indeed a distinct NIF response mostly not relying on the pulvinar and IPS. It could therefore be considered as an independent assessment of the NIF impact of light that is not dependent on the visual pathway.

The thalamus, and in particular its posterior part, seems to be crucial in the prioritization of salient stimuli[39]. This function could be achieved through changes in the excitatory/inhibitory architecture within a single thalamo-cortical loop[40]. Indeed, the thalamus seems to generate alpha oscillations, which heavily rely on inhibitory neurotransmission and might, in idling conditions, decrease cortical neuronal gain and neuronal excitability[41]. A feedback from the cortex to the thalamus can reinitiate the feedforward process[42]. In keeping with these electrophysiology findings, we found inhibitory modulations within the thalamo-cortico-thalamic loop at baseline. By contrast, the exposure to the active light transforms the thalamo-cortical connection from

inhibitory to excitatory. This shift could enhance neural communication by making it more effective, precise and selective, eventually resulting in a "wake up call" to widespread cortical territories involved in the ongoing cognitive processes[43]. Since in our design the blocks of light exposure were short and the ongoing task was easy (with a ceiling effect in performance), we do not observe a downstream improvement in performance, but we speculate that it would take place with longer exposure and/or more demanding tasks. These assumptions could be tested with a more complex network to assess whether the IPS stands in between the posterior thalamus and some other cortical areas or if the thalamus directly impact all cortical territories.

To conclude, we provided an original empirical demonstration that light affects non-visual ongoing cognitive processes at least through the modulation of the thalamo-cortical information flow in the brain. This is the first step in unraveling all the connectivity changes taking place during (blue) light exposure. Many other candidate networks could be considered, including for instance the SCN and thalamus, the LC, thalamus and cortex or the thalamus and prefrontal cortex[7]. Other factors such as light duration, spectral composition and time-of-day of the exposure warrant further investigations. Removing the difference in visual responses through the use of metameric light[44] or using human model showing a deficiency or an absence of visual function while keeping non-visual ones[45,46] could also serve to clarify the pathway linking the retina to cognition.

## Methods

**Participants.** Twenty healthy young adults were recruited. Optimal sensitivity and power analysis in MRI/DCM/PEB studies remains under investigation[47]. We nevertheless computed a sensitivity analysis to get an indication of the minimum detectable effect size in our main analysis given our sample size. According to G*Power 3 (version 3.1.9.4)[48] considering a power of 0.8, an error rate $\alpha$ of 0.25 (corrected for two tests), a sample size of 20 allowed us to detect large effect sizes $r > 0.05$ (lower limit of large effect size; 1-sided; absolute values; confidence interval: 0.07, 0.77; $R^2$ confidence interval: 0.005–0.59) within a linear multiple regression framework including 1 predictor. Based on this and on prior literature[12] we deemed the sensitivity reasonable. Our participants performed an auditory oddball task where they had to detect rare deviant tones presented pseudo-randomly within a stream of more frequent standard tones. One participant did not correctly follow the task assignment (accuracy <3 SD from the group mean). His data were therefore excluded, and nineteen participants were considered for further analysis (24.05 ± 2.63; 12 women) (Table 1). All participants provided informed consent to participate in the study and none reported a history of ophthalmic disorders. A semi-structured interview and several questionnaires assessed exclusion criteria, which were as follows: body mass index >25, clinical level of depression and anxiety, addiction or diagnosed psychiatric disorders; having worked night shift during the last year or having traveled through more than one time zone during the last 2 months, smoking, use of psychoactive drugs, excessive caffeine and alcohol consumption (i.e., >4 caffeine units/day; >14 alcohol units/week), being pregnant or at risk of pregnancy. The 21 Item Beck Anxiety[49] and Depression Inventory II[50], the Pittsburgh Sleep Quality Index[51], the Epworth Sleepiness Scale[52], the Horne-Östberg Munich chronotype questionnaire[53], and the Seasonal Pattern Assessment Questionnaire[54] were used to assess mood, sleep quality, daytime sleepiness, chronotype and changes in mood behavior respectively. The study was approved by the Ethical Committee of the University of Liège. All ethical regulations relevant to human research participants were followed. Participants were recruited

**Table 1 Characteristics of the sample.**

|  | Total sample (n = 19) |
| --- | --- |
| Age (years) | 24.05 ± 2.63 |
| BMI (kg/m$^2$) | 21.62 ± 2.74 |
| Education (years) | 14.64 ± 2.84 |
| Prior sleep duration | 7.28 ± 1.06 |
| Anxiety (BAI) | 6.05 ± 5.03 |
| Mood (BDI-II) | 6.05 ± 9.12 |
| Habitual daytime sleepiness (ESS) | 6.22 ± 3.20 |
| Chronotype (HO) | 50.5 ± 7.79 |
| Habitual sleep quality PSQI | 3.44 ± 1.82 |
| Seasonality (SPAQ) | 0.94 ± 0.72 |
| Sex | 12 F—7 M |

Characteristics of the total study sample. The education level is computed as the number of successful years of study. Scores of sleep duration as reported in the self-completion sleep evaluation questionnaire, the *BAI* (Beck Anxiety Inventory), *BDI*-II (Beck Depression Inventory II), *ESS* (Epworth Sleepiness Scale), *HO* (Horne-Ostberg), *PSQI* (Pittsburgh Sleep Quality Index) and *SPAQ* (Seasonal Pattern Assessment Questionnaire) are included. All values are provided as average ± standard deviation (*SD*).
*BMI* body mass index, *F* female, *M* male.

through advertisements on local journals and on the University of Liège website and via emails to students or members of the staff at the University. They received a financial compensation for their participation.

**Protocol and apparatus.** Participants came to the lab twice, once for a structural MRI (which also served as habituation to the experimental conditions) and then again, 7 days after, to perform a functional scan (Fig. 3). They followed a loose sleep-wake schedule (±1 h from habitual bed/wake-up time) in between scan sessions to ensure uniform circadian entrainment across participants (verified using wrist actigraphy; Cambridge Neuroscience, United Kingdom). For the functional scan, participants arrived 1.5 to 2 h after habitual wake-up time and were administered a self-completion sleep evaluation questionnaire[55] to assess aspects of sleep related to the night before. Participants were then exposed to bright polychromatic light (~1000 lux) for 5 min prior to being maintained in dim light (<10 lux) for 45 min. These light procedures were mainly implemented to standardize the participants' recent light history. Since the 19 participants included in this analysis were tested between February and November 2021, we cannot, however, exclude that light exposure during commute may have differed between subjects. A chi-square test of independence showed that there was no significant difference in the number of subjects recorded in each season ($X^2$ (3, $N = 19$) = 1.42, $p = 0.70$ ($N = 4$ in winter, $N = 6$ in spring, $N = 3$ in summer, $N = 6$ in fall). This supports that seasonal differences may have added noise to the data but are unlikely to drive the effects we report. During the light adaptation protocol, participants also received a detailed explanation of the study and were trained on the auditory tasks to be performed in the MRI. The tasks probed executive function (n-back task[56]), emotion processing[57] and attention (oddball task[58]). The present paper only discusses the later task, which was always completed following the executive task and pseudo-randomly before (8 out of 19 participants) or after (11 out of 19 participants) the emotion processing task.

For the oddball task, participants had to detect rare (20%) deviant tones (100 Hz; 500 ms) presented pseudo-randomly within a stream of more frequent (80%) standard (500 Hz; 500 ms) tones (interstimulus interval: 2 s). A short procedure preceding the task ensured optimal auditory perception of both stimuli. Acoustic stimuli were generated by a control computer located outside the MR room running OpenSesame[59] and

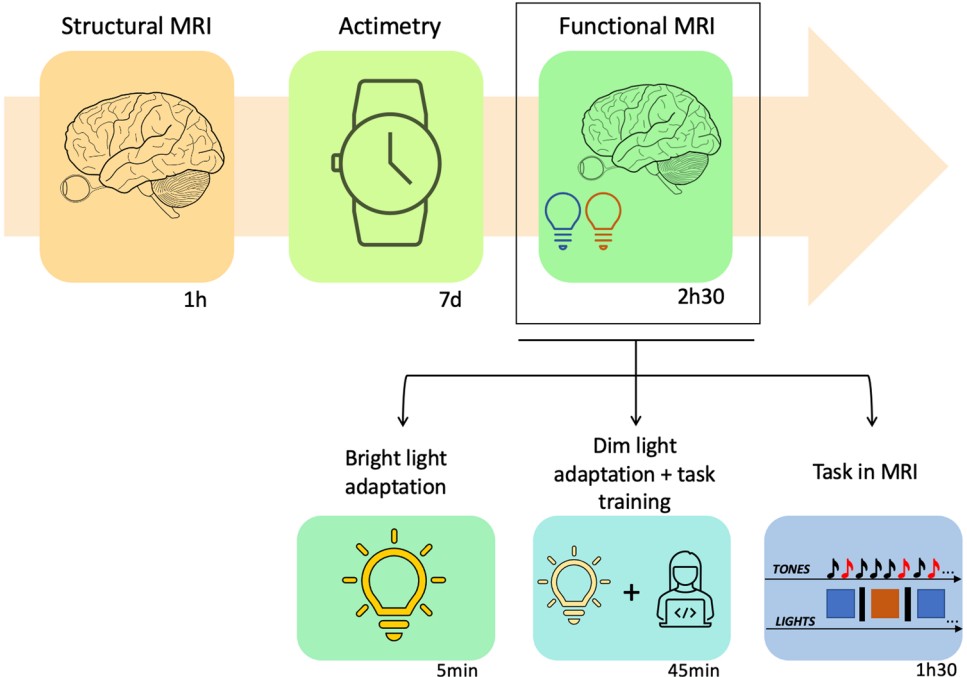

**Fig. 3 Graphical representation of the experimental protocol.** Participants came to the lab once for a structural MRI and then again, 7 days after, to perform a functional scan. During the 7 days, participants followed a loose sleep-wake schedule which was verified using wrist actigraphy. For the functional scan, participants were exposed to bright polychromatic light (~1000 lux) for 5 min prior to being maintained in dim light (<10 lux) for 45 min while they also trained for the auditory tasks to perform in the MRI. The tasks probed executive (n-back task[56]), emotion processing[57] and attention (oddball task[58]) and were performed in the MRI for about 1 h 30 min. The present paper only discusses the oddball task where participants had to detect rare (25%) deviant tones (100 Hz; 500 ms, here represented as red) presented pseudo-randomly within a stream of more frequent (75%) standard (500 Hz; 500 ms, here represented as black) tones (interstimulus interval: 2 s). While performing the task, participants were exposed to 30s-blocks of active, blue-enriched cool polychromatic light (6500 K; 92 melanopic EDI lux) or control orange monochromatic light (5.28 × 10¹² photons/cm²/s; 590 nm, 10 nm at full width half maximum; 0.16 melanopic EDI lux) separated by ~15 s darkness periods (<0.01 lux). All icons in the figure were taken from Microsoft PowerPoint (https://www.microsoft.com) except the brain icon, which was taken from Wikimedia Commons freely licensed (https://en.wikipedia.org/wiki/File:Human_Brain_sketch_with_eyes_and_cerebrellum.svg).

presented to the subjects through MR-compatible headphones (Sensimetrics, Malden, MA). Participants had to provide their responses by pressing a button on a response box (Current Designs, Philadelphia, PA) using the right index finger. While performing the task, participants were exposed to 30s-blocks of active, blue-enriched cool polychromatic light (6500 K; 92 melanopic EDI lux) meant to recruit ipRGCs photoreception, or control orange monochromatic light (5.28 × 10¹² photons/cm²/s; 590 nm, 10 nm at full width half maximum; 0.16 melanopic EDI lux) meant to trigger a visual response while recruiting much less ipRGCs. Light periods were separated by ~15 s darkness periods (<0.01 lux). The light was generated by a LED light source (SugarCUBE, Cypress, California) in the control room and transmitted to the participant using a 1-inch diameter, dual-ended optic fiber (Setra, Boxborough, Massachusetts). Switching between the active and control light conditions was accomplished by using a filter wheel (Spectral Products, AB300) and filters respectively using a UV long bypass (433–1650 nm) filter or a monochromatic orange light filter (590 nm). The dual optic fiber end was connected to a stand at the back of the head coil to illuminate the coil and ensure uniform indirect illumination of the participant's eyes. Seven blocks of each light were administered and a total of 250 standard and 63 deviant tones were delivered. The deviant tones were equally distributed across the three light blocks/conditions. During the fMRI participants' pupil size was monitored using an MRI-compatible infrared eye tracking system (EyeLink 1000Plus, SR-Research, Ottawa Canada) at a sampling rate of 1000 Hz with monocular recording

(right pupil was used). The eye tracking system returned pupil area as an arbitrary unit, i.e., the number of pixels considered part of the detected pupil. The oddball task lasted for around 12 min.

**Data acquisition.** Structural and functional MRI data were acquired using a MAGNETOM Terra 7 T MRI system (Siemens Healthineers, Erlangen, Germany) with a 32-channel receiver and 1 channel transmit head coil (Nova Medical, Wilmington, MA, USA). To improve uniformity of the $B_1$ radio frequency excitations, dielectric pads were placed between the head of the subjects and the receiver coil (Multiwave Imaging, Marseille, France). Multislice T2*-weighted fMRI images were obtained with a multi-band gradient-recalled echo—echo-planar imaging (GRE-EPI) sequence using axial slice orientation (TR = 2340 ms, TE = 24 ms, FA = 90°, no interslice gap, in-plane FoV = 224 mm × 224 mm, matrix size = 160 × 160 × 86, voxel size = 1.4 × 1.4 × 1.4 mm³). The three initial scans were discarded to avoid saturation effects. For anatomical imaging, a high-resolution T1-weighted image was acquired using a Magnetization-Prepared with 2 RApid Gradient Echoes (MP2RAGE) sequence: TR = 4300 ms, TE = 1.98 ms, FA = 5°/6°, TI = 940 ms/2830 ms, bandwidth = 240 Hz, matrix size = 256 × 256, 224 axial slices, acceleration factor = 3, voxel size = (0.75 × 0.75 × 0.75) mm³. Participants' pulse and respiration were also recoded to subsequently correct for physiological noise in the fMRI data.

**Pre-processing**. Statistical Parametric Mapping 12 (SPM12; https://www.fil.ion.ucl.ac.uk/spm/software/spm12/) under Matlab R2019 (MathWorks, Natick, Massachusetts) was used to remove high-intensity background noise, automatically reorient, and correct structural images for intensity bias. Brains were extracted to ensure optimal coregistration by using the Advanced Normalization Tools (ANTs, Penn Image Computing and Science Laboratory, UPenn, USA, http://stnava.github.io/ANTs) or RObust Brain EXtraction (ROBEX, https://www.nitrc.org/projects/robex) depending on the independent evaluation of two expert raters as to which tool yielded the best brain extraction. Brain extracted T1-images were used to create a T1-weighted group template using ANTs.

For functional volumes, voxel-displacement maps were computed using the phase and magnitude images. "Realign & Unwarp" was then applied to the EPI images to correct for head motion and for static and dynamic susceptibility induced variance[60]. Realigned and distortion corrected EPI images underwent brain extraction using the FMRIB Software Library (FSL, Analysis Group, Oxford University, UK, https://fsl.fmrib.ox.ac.uk/fsl/fslwiki)[61] and the final images were smoothed with a Gaussian kernel characterized by a full width at half maximum of 3 mm. As part of the pre-processing, we performed an additional analysis to test for differences in head motion between our active and control light condition referenced to baseline (darkness) in either the translation or the rotation axis, which showed no significant effects ($t(18) = -0.49$, $p = 0.62$, Cohen's $d = 0.18$, for translation; $t(18) = 1.06$, $p = 0.30$, Cohen's $d = 0.31$, for rotation).

For each subject, first-level analysis was performed in their native space (non-registered EPIs were used for statistical test) to prevent any possible error that may be caused by coregistration. Prior to second-level analysis, all statistical maps obtained from the first-level analysis were first transferred to the group template space and then to the MNI space ($1 \times 1 \times 1$ mm$^3$). All the registrations were performed with ANTs.

**Univariate analysis**. For each subject, changes in brain regional BOLD signal were estimated with a general linear model in which activity evoked by deviant tones in each light separately was modeled by stick functions, while blue-enriched and orange light periods were modeled by separate block functions. Sticks and blocks were convolved with a canonical hemodynamic response function. Movement parameters, as well as cardiac and respiratory parameters computed with the PhysIO Toolbox (Translational Neuromodeling Unit, ETH Zurich, Switzerland), were included as regressors of no interest. Low-frequency drifts were removed by using high-pass filtering with a cut-off period of 128 s.

The contrast of interest of the univariate analyses consisted in the main effect of deviant tones. Summary statistic images resulting from linear contrasts (in MNI space) were entered in a second-level analysis accounting for intersubject variance and corresponding to a one-sample t test for brain responses to deviant sounds. Results were corrected for multiple comparisons at the voxel level ($p < 0.05$) through a false discovery rate procedure[62].

The first principal component of BOLD signal time series was extracted bilaterally from the thalamus and IPS from the individual statistical map thresholded at $p = 0.05$ uncorrected. Individual ROI was selected from the resulting statistical map as the first cluster activated in a sphere of 8 mm radius centered on the IPS and TH coordinates extracted from the group-level univariate analysis. We also used anatomical landmarks as references for the selection of individual ROIs that were cross-checked using the Juelich Histological Atlas embedded in FSL[61].

We extracted the principal component (eigenvariate) "adjusted" time series (i.e., after regressing out effects of no interest), following the approach of Zeidman et al.[63]. BOLD time series were used to infer the underlying neural activity and were used for the effective connectivity analysis.

**Effective connectivity**. To evaluate the modulation exerted by light on the effective connectivity between thalamus and IPS during an auditory oddball task we used the Dynamic Causal Modelling (DCM)[20] framework as implemented in SPM12. Three inputs were specified in a design matrix then imported in DCM: all deviants tones trials as driving input, and the blocks of active and control light as separate modulatory inputs. Our model included mutual connections between the two regions, as reliably identified in anatomical studies in rats[64] and macaques[65], self-feedback gain control connections, and the deviant tones as reaching both regions (as both regions are not primary sensory areas receiving direct auditory information. This means that we are not specifying neither the path the stimuli follow to reach the regions nor the timing, e.g., if the regions receive the stimuli at the same time or not). The model also included the possibility that either light could exert a modulation on both connections between the thalamus and IPS.

Time series extracted from individual ROIs were carried into a first-level DCM analysis, in which our model was estimated for each subject. Then, we collapsed the DCMs for a Parametrical Empirical Bayes (PEB)[66] analysis over the first-level DCM parameter estimates. PEB is a hierarchical Bayesian model that asses commonalities and differences among subjects in the effective connectivity domain at the group level, thus taking into account the variability in the individual connections strength and reducing the weight of subjects with noisy data[67]. We carried out separate PEB analysis for each matrix (baseline connectivity, effect of driving inputs and modulatory effects) to avoid dilution of evidence effect by reducing the search space[67]. After having estimated the full model (with all connections of interest switched on) for each subject, the PEB approach performs Bayesian model reduction (BMR) and average (BMA) of the parameters across models weighted by the evidence of each model. We then used a threshold based on free energy to evaluate if a parameter contributed to the model evidence. We selected only parameters with strong evidence, meaning with posterior probability (Pp) higher than 0.95. This approach is similar to a $p$ value $\le 0.05$ in frequentist statistics (though Bayesian approach do not suffer from multiple comparison issues). Effective connectivity was estimated for the left and the right hemisphere separately.

**Pupillary response**. Pupil data was processed in Matlab R2019 (MathWorks, Natick, Massachusetts). Blinks were replaced using linear interpolation and data were smoothed using the *rlowess* robust linear regression function. Eight subjects had more than 25% of missing or corrupted pupil data and were therefore excluded from further analysis. The first 2 s of the pupil response in each light block was discarded. Data were then normalized with respect to the average pupil size during the darkness periods and averaged per light condition. To integrate both exposures to orange and blue-enriched light in a single measure we computed the difference in between normalized pupil response under orange and blue-enriched exposures (see Beckers et al.[68] for a more detailed characterization of pupillary response to light).

**Statistics and reproducibility**. Detailed explanation on how statistics were conducted on the univariate, connectivity and pupillary analysis was provided in the respective sections above.

Our experiment was not replicated. A sensitivity analysis is instead provided in the "Participants" section.

**Reporting summary**. Further information on research design is available in the Nature Portfolio Reporting Summary linked to this article.

## Data availability

The processed data are publicly available via the following open repository: https://gitlab.uliege.be/CyclotronResearchCentre/Public/fasst/dcm_light_pulvinar_ips_oddball. The raw data could be identified and linked to a single subject and represent a large amount of data. Researchers willing to access to the raw should send a request to the corresponding author (G.V.). Data sharing will require evaluation of the request by the local Research Ethics Board and the signature of a data transfer agreement (DTA).

## Code availability

The analysis scripts supporting the results included in this manuscript are publicly available via the following open repository: https://gitlab.uliege.be/CyclotronResearchCentre/Public/fasst/dcm_light_pulvinar_ips_oddball.

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

## Acknowledgements
We thank Paolo Cardone for valuable discussions and Annick Claes, Christian Degueldre, Catherine Hagelstein, Brigitte Herbillon, Patrick Hawotte, Erik Lambot, Benjamin Lauricella, André Luxen for their help over the different steps of the study. This project has received funding from the European Union's Horizon 2020 research and innovation program under the Marie Skłodowska-Curie grant agreement No 860613 (LIGHTCAP project). This study was also supported by the Belgian Fonds de la Recherche Scientifique (FRS-FNRS; CDR J.0222.20), the Fondation Léon Frédéricq, ULiège—U. Maastricht Imaging Valley, ULiège-Valeo Innovation Chair "Health and Well-Being in Transport" and Sanfran (LIGHT-CABIN project), the European Regional Development Fund (Biomed-hub), and Siemens Healthineers. None of these funding sources had any impact on the design of the study nor on the interpretation of the findings. L.L. is supported by the EU Joint Programme Neurodegenerative Disease Research (JPND) (SCAIFIELD project—FNRS reference: PINT-MULTI R.8006.20). I.C., C.P., and G.V. are supported by the FNRS. I.P. is supported by the FNRS and the GIGA Doctoral School for Health Sciences of ULiège.

## Author contributions
I.P., I.C. and G.V. designed the research. I.P., I.C., E.B., R.S. and J.F.B.A. acquired the data. A.B., E.K., N.M. and P.M. provided valuable insights while interpreting and discussing the data. C.D., P.T., L.L., S.S. and C.P. supported data acquisition by curing the MRI and fMRI sequences. I.P. analyzed the data supervised by G.V. I.P., P.M. and G.V. wrote the paper. All author edited and approved the final version of the manuscript.

## Competing interests
The authors declare no competing interests.
