## [Peer Review File · Communications Biology]

Reviewers' comments:

Reviewer #1 (Remarks to the Author):

The current work is an extension of authors' previous work examining the effect of acute light exposure on brain activity during a non-visual task. In the current study, they examined functional connectivity during the task and used DCM to specify the directionality of the thalamo-cortical connectivity. The manuscript is well-written and discussed. The major limitation is the small sample size.

I have some minor comments which I hope can help authors improve the manuscript.

1) Introduction:

-Although it is clarified/discussed later, I think it would be helpful to clarify "active" and "control" light at the very beginning. The same for explaining the rationale for measuring pupil size.

2) Results:

-please clarify why "behavioral differences would significantly bias fMRI results"?

-Did head motion differ under different light conditions or compared to baseline?

3) Methods

- in terms of recent light history, were participants tested at the same time of the year or throughout the year? The light exposure during commute might interfere with the results.

Reviewer #2 (Remarks to the Author):

Paparella et al., report on thalamo-cortical connectivity during an auditory attentional task in response to short-wavelength enriched polychromatic white light or monochromatic (orange) control light exposure. The findings maybe a meaningful contribution to the field. However, additional clarifications are warranted.

1. Is it possible to calculate post-illumination pupil response from the data collected, and if possible then can those be correlated with TH-IPS modulation as shown in Figure 2D?

2. Were any subjective measures of sleepiness/alertness collected? Were there any differences between lighting conditions? More importantly, were sleepiness ratings related to TH-IPS modulation.

3. Please report on the mean (\pm SD) sleep duration for each group.

4. Clarify what is meant by "pseudo-randomly" in "which was always completed following the executive task and pseudo-randomly before or after the emotion processing task." (Lines 281-282). How far off from random was this "pseudo-random" process and is it possible that there is a residual order effect?

5. Were participants checked for hearing or cognitive deficits?

6. Majority of the participants (12 out of 19) were females – is menstrual phase known or controlled for? Were roughly equal numbers in each lighting condition?

7. Clarify what is meant by "pseudo-randomly" in Line 308.

8. Was the short-wavelength enriched light 6500K (Line 321) or 4000K (Line 310)?

9. Figure 3 – it is unclear what the blue, red and black bars, and S's and D's refer to in the lower right box ("Task in MRI").

10. Since multiple exposures were used, can a time course profile be created?

11. Minor – change "cold" to "cool" (e.g., Line 310) when referring to color temperature.

12. Revise the last statement of the abstract "Our results provide the first empirical data supporting that blue wavelength light affects ongoing non-visual cognitive activity by modulating task-dependent information flow from subcortical to cortical areas." Although the stimulus that modulated TH-IPS information flow was indeed blue light – the implication that this may be exclusive to short-wavelength (blue) light cannot be concluded from the current study.

Reviewer 1

The current work is an extension of authors' previous work examining the effect of acute light exposure on brain activity during a non-visual task. In the current study, they examined functional connectivity during the task and used DCM to specify the directionality of the thalamo-cortical connectivity. The manuscript is well-written and discussed. The major limitation is the small sample size. I have some minor comments which I hope can help authors improve the manuscript.

We would like to thank the Reviewer for the positive evaluation of our work and for their constructive comments. We have revised the manuscript considering the Reviewer's suggestions, and hope they agree that it is improved as a result.

Minor Comments

1. *Although it is clarified/discussed later, I think it would be helpful to clarify 'active' and 'control' light at the very beginning. The same for explaining the rationale for measuring pupil size.*

We thank the Reviewer for their suggestion. We agree that this is an important clarification to make. We have added the following to the introduction to address this point (page 4, lines 47-56):

"Participants were alternatively maintained in darkness or exposed to 30s-blocks of active, blue-enriched polychromatic, or control orange monochromatic light, which, importantly, differ in terms of their predicted stimulation of ipRGCs (90 vs. 0.16 melanopic equivalent daylight illuminance -mel EDI-lux). We analyzed how the effective connectivity from the dorso-posterior thalamus to the intraparietal sulcus (IPS), a key cortical area for attentional control, during the auditory task is modulated by light information. We expected that the active light would strengthen the connectivity between the thalamus and the IPS compared to the darkness and the control light condition. We further explored, in a subset of participants, whether the impact of blue-enriched light on effective connectivity was correlated to changes in pupil size, as it constitutes another readout of the non-visual impact of light on physiology."

2. *please clarify why "behavioral differences would significantly bias fMRI results"?*

Differences in behavioral outputs must arise from differences in brain activity. This difference may, however, not be specific to the effect of interest as it could be an unspecific downstream consequence of a general modification in a brain state. As example, in the present study, a reduction in reaction times could be the results of the specific change of activity of the pulvinar or could be the results of an overall change in alertness level induced by the pulvinar. Yet, the brain must function differently to cause faster reaction time. Brain correlates of changes in reaction times could therefore be the results of specific effect of light on the pulvinar or an unspecific change in alertness level.

We slightly modified the text as follows (page 4, lines 64):

"The behavioral results are nevertheless important as they exclude that behavioral differences

unspecific to light exposure would significantly bias fMRI results.”

3. *Did head motion differ under different light conditions or compared to baseline?*

To address this question, we performed an additional analysis. We checked whether participant movement differed between the light condition. We collected movements parameters within each light condition (active, blue-enriched cold polychromatic light; and control, orange monochromatic light) as well as in the baseline darkness condition. We averaged movement absolute values separately for rotation and translation and normalized averaged movements during each light condition to the darkness period. We ran a paired t-test to test for differences across light conditions. Our analysis showed no significant difference in head motion between our active and control light condition referenced to baseline (darkness) in either the translation or the rotation axis ($t(18) = -0.49, p = 0.62$, Cohen’s $d = 0.18$, for translation; $t(18) = 1.06, p = 0.30$, Cohen’s $d = 0.31$, for rotation). Hereunder you can see the data plotted (the figure is not included in the manuscript). We now mention this additional analysis in the method section (page 17, lines 352-356):

“As part of the pre-processing, we performed an additional analysis to test for differences in head motion between our active and control light condition referenced to baseline (darkness) in either the translation or the rotation axis, which showed no significant effects ($t(18) = -0.49, p = 0.62$, Cohen’s $d = 0.18$, for translation; $t(18) = 1.06, p = 0.30$, Cohen’s $d = 0.31$, for rotation).”

Figure R1: Quantification of movements in each light condition. Quantification of participants head movements when exposed to either the active, blue-enriched polychromatic, or the control, orange monochromatic, light condition referenced to baseline (darkness). Head movements are reported separately for the rotation or the translation axis. Paired t-test did not show any significant effect ($t(18) = -0.49, p = 0.62$, Cohen’s $d = 0.18$, for translation; $t(18) = 1.06, p = 0.30$, Cohen’s $d = 0.31$, for rotation) of the light condition on the head movements. Individual data are shown. Error bars represent ± 1 within-subjects Standard Error of the Mean (SEM) (Cousineau, 2005).

4. *in terms of recent light history, were participants tested at the same time of the year or throughout the year? The light exposure during commute might interfere with the results.*

Thank you for highlighting this. We included an adaptation light protocol before the fMRI, where participants were exposed to bright polychromatic light (~1000 lux) for 5 minutes prior to being maintained in dim light (< 10 lux) for 45 min. This procedure was implemented to standardize participants' recent light history. Since the 19 participants included in this analysis were tested between February and November 2021, we cannot, however, exclude that light exposure during commute may have differed between subjects. A chi-square test of independence showed that there was no significant difference in the number of subjects recorded in each season ($\chi^2(3, N = 19) = 1.42, p = 0.70$ (N = 4 in winter, N = 6 in spring, N = 3 in summer, N = 6 in fall)). This supports that seasonal differences may have added noise to the data but are unlikely to drive the effects we report." We now report this in the method section (page 13, lines 262-270):

"These light procedures were **mainly** implemented to standardize the participants' recent light history. **Since the 19 participants included in this analysis were tested between February and November 2021, we cannot, however, exclude that light exposure during commute may have differed between subjects. A chi-square test of independence showed that there was no significant difference in the number of subjects recorded in each season ($\chi^2(3, N = 19) = 1.42, p = 0.70$ (N = 4 in winter, N = 6 in spring, N = 3 in summer, N = 6 in fall)). This supports that seasonal differences may have added noise to the data but are unlikely to drive the effects we report. During the light adaptation protocol, participants also received** a detailed explanation of the study and **trained** on the auditory tasks to be performed in the MRI".

Reviewer 2

Paparella et al., report on thalamo-cortical connectivity during an auditory attentional task in response to short-wavelength enriched polychromatic white light or monochromatic (orange) control light exposure. The findings maybe a meaningful contribution to the field. However, additional clarifications are warranted.

We would like to thank the Reviewer for making helpful comments which have given us a chance to clarify important aspects of our work. Please find below the responses to each of the point raised by the Reviewer.

Major Comments

1. *Is it possible to calculate post-illumination pupil response from the data collected, and if possible then can those be correlated with TH-IPS modulation as shown in Figure 2D?*

While we see the potential of computing this additional measure and test its link with TH-IPS modulation, we did not compute post-illumination pupil response on our subsample of 11 participants for several reasons. We are mainly interested in the non-visual effects of light during exposure. That is why we analyzed pupil responses during our active vs. control light exposure as normalized to the darkness condition. Moreover, in our protocol, while doing the oddball task participants were exposed to 30s-blocks of active, or control light, which were separated by ~15s darkness periods. We thus believe that the shortness of our dark periods and their variation in length could affect the reliability of a post-illumination pupil response measure. Finally, we decided to analyze the pupillary response of a subsample of participants as an exploratory analysis to complement our main findings showing that the TH-IPS modulation exerted by blue-enriched light relates to a non-visual response to light. We hope the Reviewer sees our explanations as reasonable and agrees that including further analysis on the pupil data would fall outside the scope of our manuscript.

2. *Were any subjective measures of sleepiness/alertness collected? Were there any differences between lighting conditions? More importantly, were sleepiness ratings related to TH-IPS modulation.*

We thank the Reviewer for this important question, which gives us the chance to clarify our protocol. Participants' sleepiness was assessed orally before and after they completed the fMRI tasks using the Karolinska Sleepiness Scale (KSS) (Åkerstedt & Gillberg, 1990). Unfortunately, given our protocol, we could not assess participants sleepiness/alertness before and after the oddball tasks neither could we do it after each block of light exposure. We therefore do not have the data to test eventual differences between light conditions as requested by the Reviewer.

In an attempt to satisfy the reviewer's request, we nevertheless tested whether participants' overall level of sleepiness/alertness was related to the TH-IPS modulation exerted by blue-enriched light we observe. We computed 3 different Pearson's correlation analysis, to test whether participants'

alertness before ($r[9] = -0.12$; $p = 0.61$), after ($r[9] = 0.07$; $p = 0.78$) or after light exposure as compared to the baseline (before light exposure) ($r[9] = 0.38$; $p = 0.12$) were correlated to the TH-IPS modulation, and all showed no significant correlation. For the sake of clarity and simplicity of the manuscript to the reader we did not include these analyses in the revised manuscript.

REF:

Åkerstedt, T. & Gillberg, M. Subjective and objective sleepiness in the active individual. *Int. J. Neurosci.* 52, 29–37 (1990)

Figure R2: Association between Thalamus-IPS connectivity changes and subjective sleepiness collected before and after the MRI protocol. Three separate Pearson’s correlation analysis to test whether participants’ alertness before ($r[9] = -0.12$; $p = 0.61$), after ($r[9] = 0.07$; $p = 0.78$) or after light exposure as compared to the baseline (before light exposure) ($r[9] = 0.38$; $p = 0.12$) were correlated to the TH-IPS modulation. All showed no significant correlation.

3. Please report on the mean (\pm SD) sleep duration for each group.

We thank the Reviewer for this suggestion. We recorded self-reported sleep duration according to a self-completion sleep evaluation questionnaire (Parrott & Hindmarch, 1978) and double-checked it with what was reported in the sleep diary participants were asked to fill before coming for the fMRI session. Sleep duration was added in Table 1 which includes all the characteristics of our sample (page 14, lines 276-283). We also spelled out the other variables for clarity.

	Total sample (n = 19)
Age (years)	24.05 ± 2.63
BMI (kg/m ²)	21.62 ± 2.74
Education (years)	14.64 ± 2.84
Prior sleep duration	7.28 ± 1.06
Anxiety (BAI)	6.05 ± 5.03
Mood (BDI-II)	6.05 ± 9.12
Habitual daytime sleepiness (ESS)	6.22 ± 3.20
Chronotype (HO)	50.5 ± 7.79
Habitual sleep quality PSQI	3.44 ± 1.82
Seasonality (SPAQ)	0.94 ± 0.72
Sex	12F – 7 M

Table 1. Characteristics of the sample.

Characteristics of the total study sample. BMI: Body Mass Index. The education level is computed as the number of successful years of study. Scores of **sleep duration as reported in the self-completion sleep evaluation questionnaire**, the BAI (Beck Anxiety Inventory), BDI-II (Beck Depression Inventory II), ESS (Epworth Sleepiness Scale), HO (Horne-Ostberg), PSQI (Pittsburgh Sleep Quality Index) and SPAQ (Seasonal Pattern Assessment Questionnaire) are included. All values are provided as average ± standard deviation (SD). F: Female, M: Male.

REF:

Parrott, A. C., & Hindmarch, I. (1978). Factor analysis of a sleep evaluation questionnaire. *Psychological medicine*, 8(2), 325-329.

4. Clarify what is meant by “pseudo-randomly” in “which was always completed following the executive task and pseudo-randomly before or after the emotion processing task.” (Lines 281-

282). *How far off from random was this “pseudo-random” process and is it possible that there is a residual order effect?*

The oddball task, discussed in this manuscript, was always completed after an executive task and either before or after an emotion processing task. 8 of the 19 participants included in our manuscript completed the oddball task before and 11 after the emotion task. This information has been also included in the revised manuscript (page 13, lines 272-274).

“The present paper only discusses the later task, which was always completed following the executive task and pseudo-randomly before (8 out of 19 participants) or after (11 out of 19 participants) the emotion processing task”.

5. *Were participants checked for hearing or cognitive deficits?*

No participant reported cognitive or hearing problem. Given the age of our sample (24.05 ± 2.63) we did not run any hearing test nor any cognitive test (that are only sensitive to marked age-related cognitive decline). A short volume-check procedure preceding the task while participants were already lying in the MRI scanner ensured optimal perception of the auditory stimuli.

6. *Majority of the participants (12 out of 19) were females – is menstrual phase known or controlled for? Were roughly equal numbers in each lighting condition?*

Unfortunately, we did not collect female participants’ menstrual phase. As ours is a within-subjects experimental design we did not think we had to account or control for this.

7. *Clarify what is meant by “pseudo-randomly” in Line 308?*

In the oddball task, participants had to detect rare deviant tones among more frequent standard ones. Deviant tones were presented pseudo-randomly to ensure their spread over the entire duration of the task. Deviant tones were not presented in a truly random fashion as we needed to make sure their presentation was balanced across different light blocks and across the entire task. Seven blocks of each light were administered and a total of 250 standard and 63 deviant tones were delivered. The deviant tones were equally distributed across the three light conditions (active, blue-enriched cold polychromatic light; control, orange monochromatic light; darkness).

8. *Was the short-wavelength enriched light 6500K (Line 321) or 4000K (Line 310)?*

We thank the Reviewer for spotting this error. The short wavelength enriched light is indeed 6500K. The error has been corrected in the revised manuscript (page 15, lines 293-296):

“While performing the task, participants were exposed to 30s-blocks of active, blue-enriched cold polychromatic light (6500K; 92 melanopic EDI lux) or control orange monochromatic light (5.28×10^{12} photons/cm²/s; 590nm, 10nm at full width half maximum; 0.16 melanopic EDI lux) separated by ~15s darkness periods (<0.01 lux)”.

9. *Figure 3 – it is unclear what the blue, red and black bars, and S’s and D’s refer to in the lower right box (“Task in MRI”).*

Following the Reviewer’s comment, we have slightly modified Figure 3 and its caption. We represent tones of different colors to differentiate between deviant (red) and standard (black) tones. We also added some text to indicate that the blue, orange and black bars refer to the light blocks participants were exposed to when doing the task. The revised figure appears also in the revised manuscript (page 15, lines 285-296).

Figure 3. Graphical representation of the experimental protocol. Participants came to the lab once for a

structural MRI and then again, 7 days after, to perform a functional scan. During the 7 days, participants followed a loose sleep-wake schedule which was verified using wrist actigraphy. For the functional scan, participants were exposed to bright polychromatic light (~1000 lux) for 5 minutes prior to being maintained in dim light (< 10 lux) for 45 min while they also trained for the auditory tasks to perform in the MRI. The tasks probed executive (n-back task⁵³), emotion processing⁵⁴ and attention (oddball task⁵⁵) and were performed in the MRI for about 1h 30min. The present paper only discusses the oddball task where participants had to detect rare (25%) deviant tones (100Hz; 500ms, here represented as red) presented pseudo-randomly within a stream of more frequent (75%) standard (500Hz; 500ms, here represented as black) tones (interstimulus interval: 2s). While performing the task, participants were exposed to 30s-blocks of active, blue-enriched cool polychromatic light (6500K; 92 melanopic EDI lux) or control orange monochromatic light (5.28x10¹²photons/cm²/s; 590nm, 10nm at full width half maximum; 0.16 melanopic EDI lux) separated by ~15s darkness periods (<0.01 lux).

10. *Since multiple exposures were used, can a time course profile be created?*

The repetition of light blocks within a session ensures good regression of any potential effects of interest in the statistical analyses. Our focus was to assess robust connectivity changes over the entire tasks, i.e. beyond any other changes such as time effect. Given the relatively low number of participants we did not consider adding time as modulator in our model.

Minor Comments

1. *Minor – change “cold” to “cool” (e.g., Line 310) when referring to color temperature.*

We thank the Reviewer for highlighting this. “Cold” has been replaced with “cool” throughout the revised manuscript, when referring to color temperature.

2. *Revise the last statement of the abstract “Our results provide the first empirical data supporting that blue wavelength light affects ongoing non-visual cognitive activity by modulating task-dependent information flow from subcortical to cortical areas.” Although the stimulus that modulated TH-IPS information flow was indeed blue light – the implication that this may be exclusive to short-wavelength (blue) light cannot be concluded from the current study*

Following the Reviewer’s comment, we have modified the abstract to avoid implying that our results are exclusive to blue wavelength light. The revised abstract now reads (page 2, lines 10-12):

“Our results provide the first empirical data supporting that blue-wavelength light affects ongoing non-visual cognitive activity by modulating task-dependent information flow from subcortical to cortical areas”.

REVIEWERS' COMMENTS:

Reviewer #1 (Remarks to the Author):

Authors have addressed most of my concerns.
However I don't agree on the response that lack of behavioral differences suggests that there was no difference in alertness level. Authors' response to Reviewer2' second comment might be a better argument. I would simply delete the sentence.

Reviewer #2 (Remarks to the Author):

The the clarifications are helpful.

Reviewer 1

Authors have addressed most of my concerns. However, I don't agree on the response that lack of behavioral differences suggests that there was no difference in alertness level. Authors' response to Reviewer2' second comment might be a better argument. I would simply delete the sentence.

We would like to thank the Reviewer for their suggestion. However, we are still convinced that the statement is important and should be included in the text.

We modified the text as follows to take into account the relevant comment of the Reviewer (PAGE 4):

“The behavioral results are nevertheless important as they exclude that behavioral performance differences unspecific to light exposure would significantly bias fMRI results. The measures we collected during the protocol does not allow us to exclude however that alertness and/or attention were affected by the short light exposure without affecting performance to the task (see Campbell et al. 2023 for an impact of light on evoked pupil responses that are related to attention and alertness).”

Campbell, I. *et al.* Impact of light on task-evoked pupil responses during cognitive tasks. *bioRxiv* 2004–2023 (2023).

Reviewer 2

The clarifications are helpful.

We would like to thank the Reviewer for making helpful comments and for their suggestions, which we believe have significantly improved our manuscript.